# Metabolomic Profile of BALB/c Macrophages Infected with *Leishmania amazonensis*: Deciphering L-Arginine Metabolism

**DOI:** 10.3390/ijms20246248

**Published:** 2019-12-11

**Authors:** Sandra Marcia Muxel, Maricruz Mamani-Huanca, Juliana Ide Aoki, Ricardo Andrade Zampieri, Lucile Maria Floeter-Winter, Ángeles López-Gonzálvez, Coral Barbas

**Affiliations:** 1Departamento de Fisiologia, Instituto de Biociências, Universidade de São Paulo, São Paulo, SP 05508-090, Brazil; juaoki@usp.br (J.I.A.); ricardoz@ib.usp.br (R.A.Z.); lucile@usp.br (L.M.F.-W.); 2Centre for Metabolomics and Bioanalysis (CEMBIO), Departamento de Química e Bioquímica, Facultad de Farmacia, Universidad CEU San Pablo, Campus Monteprincipe, Boadilla del Monte, 28660 Madrid, Spain; maricruzmh31@gmail.com (M.M.-H.); cbarbas@ceu.es (C.B.)

**Keywords:** polyamines pathway, metabolites, bone marrow-derived macrophages, arginine

## Abstract

Background: Leishmaniases are neglected tropical diseases that are caused by *Leishmania*, being endemic worldwide. L-arginine is an essential amino acid that is required for polyamines production on mammal cells. During *Leishmania* infection of macrophages, L-arginine is used by host and parasite arginase to produce polyamines, leading to parasite survival; or, by nitric oxide synthase 2 to produce nitric oxide leading to parasite killing. Here, we determined the metabolomic profile of BALB/c macrophages that were infected with *L. amazonensis* wild type or with *L. amazonensis* arginase knockout, correlating the regulation of L-arginine metabolism from both host and parasite. Methods: The metabolites of infected macrophages were analyzed by capillary electrophoresis coupled with mass spectrometry (CE-MS). The metabolic fingerprints analysis provided the dual profile from the host and parasite. Results: We observed increased levels of proline, glutamic acid, glutamine, L-arginine, ornithine, and putrescine in infected-*L. amazonensis* wild type macrophages, which indicated that this infection induces the polyamine production. Despite this, we observed reduced levels of ornithine, proline, and trypanothione in infected-*L. amazonensis* arginase knockout macrophages, indicating that this infection reduces the polyamine production. Conclusions: The metabolome fingerprint indicated that *Leishmania* infection alters the L-arginine/polyamines/trypanothione metabolism inside the host cell and the parasite arginase impacts on L-arginine metabolism and polyamine production, defining the infection fate.

## 1. Introduction

Cutaneous, mucocutaneous, or visceral manifestations characterize *Leishmania* infection. Leishmaniases are neglected tropical diseases, being endemic around the world, and affecting more than 12 million people, with an annual incidence estimated to be 1.5 million cases for cutaneous manifestations and 300,000 cases for visceral manifestations. *Leishmania amazonensis* causes cutaneous and/or diffuse cutaneous manifestations [1,2]. *Leishmania* is a protozoan parasite that alternates its life cycle between the sand fly and mammalian hosts. The parasite can sense differences in the environment and rapidly modulate gene expression and cellular signaling, inducing differentiation from metacyclic promastigote forms found in the proboscis of the phlebotomine sand fly host, to amastigote forms in the interior of macrophage phagolysosomes in the mammalian host [3]. *Leishmania* infection results in the regulation of pathways that are involved in the inflammatory response and leishmanicidal mechanisms, such as nitric oxide (NO) production via nitric oxide synthase 2 (NOS2), which is induced by Th1-cytokines [4,5,6,7,8,9]. However, *Leishmania* can subvert these mechanisms, which impacts on Th1/Th2 cytokine balance and induces macrophage arginase 1 (ARG1) activity to produce polyamines, putrescine, spermidine, and spermine, leading to *Leishmania* survival [6,10,11,12]. 

L-arginine is an essential amino acid and a precursor in the synthesis of proteins, urea, ornithine, citrulline, NO, creatinine, agmatine, glutamate, proline, putrescine, spermidine, and spermine, supporting the proline, glutamate, and polyamine metabolism at the whole organism level or cellular level in mammals. Its availability and metabolization can modulate inflammation and immune response regulation during infections and also recover the physiological steady-state [13,14].

In mammalian cells, the endogenous synthesis of L-arginine from proline or glutamate via ornithine is not sufficient for supplying several pathways that need this amino acid as a precursor. L-arginine uptakes occur via cationic amino acid transporters (CAT1, CAT2A, CAT2B, and CAT3) to supply the metabolic pathways. CAT transporters have a different affinity to L-arginine, ornithine, histidine, and lysine [15,16]. The recognition of pathogen signals mediates activation and proliferation of macrophages promoting an augment of CAT2 levels and L-arginine uptake [17]. In *Leishmania*-infected cells, L-arginine uptakes occur via CAT1 and CAT2 to supply the requirement for host-enzymes NOS2 and ARG1 [18,19,20]. In contrast, L-arginine uptake in *Leishmania* promastigote and amastigote forms occurs via amino acid permease 3 (AAP3) to supply parasite arginase (ARG) or to NOS-like, to produce polyamines or NO, respectively, which leads to parasite growth [18,21,22,23]. Amastigote that forms inside the host cell can scavenge the amino acid and other nutrients from phagolysosome, thus interfering in the host cell metabolism [24,25,26]. The availability of L-arginine establishes a competition between the parasite and macrophage enzymes, altering gene expression regulation and, consequently, NO/polyamines production, as well as macrophage susceptibility to infection [18,21,22,23]. Previous studies demonstrated that the metabolic fingerprint analysis of *L. amazonensis* wild type (La-WT) promastigote forms under the starvation of L-arginine and revealed decreased levels of ornithine and putrescine, whereas it did not alter the levels of spermidine, spermine, or agmatine [27]. Besides, the analysis of *L. amazonensis* arginase knockout (La-arg^−^) promastigote forms revealed increased levels of L-arginine and citrulline, but it decreased levels of ornithine and putrescine [27,28,29]. 

The whole sequencing of genome and transcriptome approach and also the metabolome profiles from parasite and host cells have helped in identifying the conserved metabolic pathways and the perturbation on the homeostasis conditions during infections, pointing to potential drug targets [30,31,32,33,34]. Based on that, in this work, we analyzed the dual metabolomic profiles metabolomics of BALB/c macrophages infected with La-WT or La-arg^−^. The data presented here focused on L-arginine metabolism, which can help in the understanding of the complexity of host-parasite interactions, and thereby supporting the search for new drug targets to control *Leishmania* outcome. We showed the global impact on the metabolic fingerprint of macrophages during La-WT infection, pointing to the regulation of L-arginine metabolism and polyamine biosynthesis, by increased levels of proline, glutamic acid, glutamine, L-arginine, ornithine, and putrescine, while using the Capillary Electrophoresis-Mass Spectrometry (CE-MS) approach. The absence of parasite arginase activity distinctly impacted on macrophage metabolism, increasing the L-arginine levels, but reducing the ornithine, proline, and trypanothione levels. Altogether, these data indicate a differentiation in the metabolomics profile among uninfected macrophages, La-WT or La-arg^−^ infected macrophages.

## 2. Results

### 2.1. L. amazonensis Infected Macrophages Show a Differential Metabolomic Profile than Uninfected Macrophages

Macrophages were infected with La-WT or La-arg^−^ promastigote forms in the stationary growth phase (MOI 5:1) and we analyzed the number of infected macrophages, amastigotes per macrophage, and the infectivity index (the mean number of amastigotes per macrophage multiplied by the rate of macrophage infection). According to Appendix A, we observed that the percentage of infected macrophages was 28% for La-WT and 23% for La-arg^−^. Indeed, the number of amastigotes per infected macrophage was 2.8 for La-WT and 2.4 for La-arg^−^. The infectivity index was 1.1 for La-WT and 0.75 for La-arg^−^-infected macrophages. No statistical differences were observed in this early time of infection. 

Once these results were corroborated with previous data, observed for both peritoneal [29] and BMDMs [23] infections, metabolite extraction, and CE-MS analysis were performed. First, we determined the PCA-X scores plot from samples as compared to Quality-control (QC), to check the stability and reproducibility of the CE-MS system. The QCs, randomly distributed throughout the worklist, should be grouped in the center of the graph, showing high stability and reproducibility of the system throughout the analysis. Afterwards, the QCs were deleted and choosing only those data present in at least one treatment filtered the remaining data; Appendix A shows this model. The QC analysis showed *R*^2^ = 0.951, indicating a good quality of samples. Additionally, the samples were grouped in three distinct clusters, represented by uninfected macrophages, La-WT-infected macrophages, and La-arg^−^ infected macrophages. Interestingly, the La-arg^−^-infected macrophages cluster appeared to be closer to the uninfected macrophages than the La-WT-infected macrophages cluster, which indicated that the absence of parasite arginase impacted on the metabolome profile (Appendix A).

After this initial evaluation, the orthogonal partial least-squares-discriminant analysis (OPLS-DA) models were generated for comparisons between two groups at a time: (**A**): uninfected macrophages and La-WT-infected macrophages, (**B**) uninfected macrophages and La-arg^−^-infected macrophages, and (**C**) La-WT-infected macrophages and La-arg^−^-infected macrophages, to select the metabolites that are responsible for sample clustering and separation of the groups (Figure 1). In all cases, the models presented good quality, being *R*^2^ ≥ 0.945 and Q^2^ ≥ 0.764, respectively, Jack-Knife confidence intervals were calculated to identify the features that were statistically significant (*p* correlation (*p* corr) > |0.5| and Variable Importance in Projection (VIP) > 1).

Through the chemometric tests (ANOVA with FDR-corrected *p* value) and the Jack-Knife confidence intervals, the characteristics that showed a statistically significant difference (*p* value < 0.05) between the treatments were selected. These characteristics were identified by their exact mass and RMT while using an in-house database (http://ceumass.eps.uspceu.es/mediator//) (Table 1). The metabolome profiles of samples shown in Table 1 and Appendix A represent a mixture of metabolites from both parasite and host cells since metabolites and metabolic pathways in parasites and host cells are conserved [35]. A total of 135 features were found in uninfected macrophage samples, while 163 were found in La-WT-infected and La-arg^−^-infected macrophages. Further, the statistical metabolite abundance analysis among the comparisons was performed and revealed the modulation of 65 metabolites (Table 1). From those, 58 metabolites appeared to be modulated in La-WT-infected as compared to uninfected macrophages, which indicated that *L. amazonensis* infection impacted on metabolite abundance, which can be mediated by its own metabolism inside macrophages and/or manipulate the host metabolism. Indeed, 65 metabolites appeared to be modified in La-arg^−^-infected as compared to uninfected macrophages, as a consequence of parasite metabolism into the host cell and macrophage activation of microbicide metabolism (Table 1).

Additionally, we designed a heat map comprising the replicate samples of the comparisons (Figure 2). We observed a distinct cluster of La-WT-infected macrophages and La-arg^−^-infected macrophages in comparison with uninfected macrophages, according to the figure. 

Interestingly, only 20 metabolites appeared to be modulated in the comparison of La-arg^−^-infected and La-WT-infected macrophages (Table 2). These data indicated that the absence of parasite arginase activity impacted the metabolite concentrations. 

### 2.2. L. amazonensis Infection of Macrophages Impact on the Modulation of Arginine and Proline Metabolism

We evaluated the significant changes of metabolites and the correlation with the metabolic pathways, while using an enrichment analysis from MetaboloAnalyst 4.0 Software, based on metabolites peak areas from La-WT-infected as compared to La-arg^−^-infected macrophages (Figure 3, Appendix A), La-WT-infected compared to uninfected macrophages (Appendix A), and La-arg^−^-infected macrophages when compared to uninfected macrophages (Appendix A) to understand how infection alters macrophage metabolism. When compared La-WT-infected and uninfected macrophages (Appendix A), we observed a modulation on a trehalose degradation, nucleotide sugars metabolism, starch and sucrose metabolism, and gluconeogenesis, and also in the urea cycle. In the same way, the comparison of La-arg^−^-infected macrophages and uninfected macrophages (Appendix A), as shown in the modulation of nucleotide sugars metabolism, starch and sucrose metabolism, gluconeogenesis and galactose metabolism, and also in urea cycle. These data suggest an increase in the recruitment of energy sources. 

Interestingly, the following pathways revealed a higher modulation in the comparison of La-arg^−^-infected versus La-WT-infected macrophages: arginine and proline metabolism, urea cycle, beta-alanine metabolism, and spermidine and spermine biosynthesis (Figure 3, Appendix A). These findings indicated a distinct modulation of arginine and proline metabolism during *L. amazonensis* infection, which strengthens parasite arginase activity and its role in host metabolism.

### 2.3. Parasite Metabolism Intercross the Changes in L-Arginine and Proline Metabolism During Infection 

While focusing on the arginine and proline metabolism altered in infected macrophages (La-WT and La-arg^−^) as compared to uninfected macrophages, we intended to decipher this pathway through these comparisons. The L-arginine metabolism starts with the metabolizing of this amino acid to citrulline, NO via NOS2 or to urea, and ornithine via arginase; subsequently, ornithine is converted in putrescine, supplying polyamine metabolism, or to proline, contributing to form glutamate (Figure 4A).

According to the profile of metabolite content, we observed increased levels of proline, *trans*-4-hidroxiproline, glutamic acid, glutamine, L-arginine, ornithine, putrescine, gamma-glutamyl-L-cysteine, glutathionylspermidine, trypanothione, trypanothione disulfide, S-adenosylmethionine, 5’-methylthioadenosine, methionine, cystathionine, L-alanyl-aspartate, and glycine in La-WT-infection. The spermidine, argininic acid, and citrulline levels appeared to be maintained (Figure 4B; Table 1). The absence of parasite arginase distinctly increased the levels of L-arginine, citrulline, and argininic acid, but reduced the levels of ornithine, proline, and trypanothione (Figure 4B, Table 1 and Table 2). These data corroborated that infection deviates L-arginine metabolism to polyamine production, and also parasite arginase activity cooperates in the balance of host metabolism. 

## 3. Discussion

Macrophages are cells that are specialized to recognize and phagocytize harmful organisms, including the *Leishmania* parasite, and induce microbicidal activity. However, the parasite can subvert the activation of macrophages to establish infection [36]. The genomic and transcriptomic networks during *Leishmania* infection have been providing interesting data regarding how *Leishmania* can to modulate the gene organization and gene expression of its host [37,38,39,40]. Additionally, we have observed differentially expressed genes profile during the different phases of parasite growth [37,38,41]. The metabolomic profile is poorly explored and it could help to understand the metabolite’s consummation and network systems pathway of both the host and parasite. The previous study detailed the parasite metabolic profile showing that La-WT under L-arginine deprivation decreases the levels of ornithine and putrescine, but it did not alter the levels of spermidine, spermine, or agmatine [27]. In contrast, the La-arg^−^ promastigote forms showed increased levels of L-arginine and citrulline, but decreased levels of ornithine and putrescine [27]. Here, we described the dual metabolites profile from BALB/c macrophages that were infected with La-WT or La-arg^−^ during the early time of infection and focused on the modulation of L-arginine and proline metabolism.

*Leishmania* is auxotrophic for many amino acids, including L-arginine, proline, and glutamine, presenting complex machinery to uptake the exogenous source that is necessary for its replication and growth [42,43]. Indeed, intracellular amastigotes can scavenge sugar, lipids, and amino acids, such as L-arginine, from the host phagolysosome [24,26]. L-arginine availability and arginase activity play important roles in the gene expression and metabolites profile from L-arginine and polyamines metabolism in *Leishmani**a* promastigote and amastigote forms and it also impacts on parasite survival and growth [22,27,28]. In the host context, the macrophage ARG1 activity can be activated to provide substrates for polyamine pathways, and the parasite shares the L-arginine and polyamine intracellular pool with the macrophage [36,44]. Additionally, the depletion of L-arginine in the phagolysosome by amastigotes alters the cytoplasmic pool of L-arginine and results in a reduction of NO levels, leading to susceptibility to *L. major*, *L. amazonensis*, and *L. donovani* infection in BALB/c mice [6,10,11,12,45,46]. On the other hand, the higher expression of NOS2 deviates the use of L-arginine to produce NO in infected macrophages, promoting parasite killing [43,47], and enabling resistance to infection in C57BL/6 mice infection [4,5,6,7,8,48]. 

La-WT infection in BALB/c-macrophages increases the level of L-arginine via the increased expression of *Cat*2 and *Cat*1 and the uptake of amino acids as compared to uninfected macrophages [23]. Besides, the levels of La-*aap*3 expression also appeared to be increased, but did not increase in BALB/c-macrophages that were infected with La-arg^−^ [22,23]. The metabolomic profile in La-WT infection showed augmented levels of glucose and glutamine, which can be implicated in the activation of macrophages and glycolytic metabolism to generate ATP via aerobic metabolism in pro-inflammatory M1 macrophages, which reduces the oxidative phosphorylation (OXPHOS) and fatty acid oxidation (FAO) pathways that surpass two points of tricarboxylic acid cycle (TCA) by supplying glutamine and acetyl-CoA, respectively; in contrast, alternatively activated M2 macrophages that display a more flexible metabolic activity since they increase OXPHOS, and increase the production of polyamines for cell proliferation or proline to induce collagen production [49]. The levels of the polyamines ornithine and putrescine also increased, which correlated with the increased levels of *Arg*1 mRNA and parasite ARG [23] directing to the production of ornithine and putrescine via ODC [23,37] (Figure 4 and Figure 5). Additionally, the increased levels of ornithine correlate with increased levels of proline, *trans*-4-hidroxiproline, glutamic acid, and glutamine. The increased levels of citrulline can be a consequence of citrulline production via L-arginine conversion by host arginine deaminase, or ornithine conversion by ornithine transcarbamilase (OTC) [50]; subsequently, the infection reduces *Nos2* mRNA and NOS2 protein levels and NO production [23]. The NO production in the initial steps of the inflammatory response could help the leishmanicidal activity, as observed for *L. major* infection of BALB/c and C57BL/6 mice [51], although the parasite can surpass this response, surviving and replicating inside the macrophages [36,52,53,54].

In contrast, BALB/c infection with the La-arg^−^ parasite showed increased levels of L-arginine, supporting the uptake of amino acids by increased levels of *Cat*2 and *Cat*1 mRNA expression [23]. Excess L-arginine availability to the macrophages, which was not metabolized via arginase (absent in parasite) or agmatinase that was not modulated in amastigotes axenic forms, could cause the presence of higher levels of L-argininic acid [37]. Additionally, the absence of parasite arginase reduced the levels of ornithine (as compared to La-WT infected macrophages), although the increased levels of *Arg*1 in infected La-arg^−^-macrophages when compared to infected with La-WT [23], highlighting the impact of parasite arginase into the total of L-arginine conversion to ornithine. Additionally, in the absence of parasite arginase, L-arginine can be used by NOS2 for NO and citrulline production, which are increased in La-arg^−^-macrophages [23], deviating L-arginine from host-ARG1. In the same way, the absence of arginase from *L. amazonensis* and *L. mexicana* increases the NO levels and reduces infectivity in macrophages [23,29,55]. Indeed, the enzymatic kinetics of host-NOS2 and ARG1 can be considered, once the substrate concentration for the enzymatic kinetics of ARG1 (k*m* of 1 mmol/L) is higher than NOS2 (k*m* of 100 μmol/L), interfering with the balance of L-arginine use in our context [56,57].

In La-arg^−^-macrophages, the lower levels of ornithine and proline can correlate, once the levels of ornithine cyclodeaminase, which is an enzyme that converts ornithine in proline, do not modify in La-arg^−^-amastigote axenic forms as compared to La-WT [37]. Besides, lower levels of ornithine also can reduce their conversion to L-glutamate 5-semihaldehyde and subsequently in glutamate or L-1-pyrroline 5-carboxylate, metabolites that were not differentially modulated in comparison to La-arg^−^-macrophages versus La-WT-infected.

Glutathione is the main thiol antioxidant protecting mammalian cells against oxidative stress that is induced by oxygen- and nitrogen-derived reactive species, acting as a ROS scavenger in thiol-disulfide reactions to control the protein functions. Indeed, we observed increased levels of ϒ -L-Glutamyl-L-cysteine and reduced levels of S-nitroso-glutathione, but glutathione was not modulated in infected macrophages [58]. Further, the spermidine level was not modulated in infected macrophages, which could be a result of its use by glutationylspermidine synthase for conversion in glutathionylspermidine, while using glutathione [58], and subsequently in trypanothione and trypanothione disulfide. Trypanosomes, including *Leishmania*, produce trypanothione to protect the parasite against the mammalian host defense, acting as an intracellular thiol redox balance [59]. *L. infantum*, antimonial-resistant strain, was able to modify the L-arginine metabolism and polyamine pathways by guiding to the thiol-dependent redox metabolism [60]. Indeed, infection reduced the level of S-nitroso-glutathione, which is a nitrosothiol that releases NO and mimics the effects of endogenous NO and nitrosylation of protein thiols, which implicates NO-dependent regulation of many enzymes, including glyceraldehyde-3-phosphate dehydrogenase [61] and energy metabolism-related enzymes that are involved in glycolysis, and also interferes in the inflammatory response of macrophages [62].

In conclusion, the metabolomic profile that is presented here represents the dual metabolite pattern from host and parasite, where L-arginine/polyamines/trypanothione metabolism showed higher modulation during *L. amazonensis* infection, leading to polyamine production and intracellular thiol redox balance and, consequently, the survival of parasite inside the host-cell (Figure 5). The activity of parasite arginase cooperated with a deviation of L-arginine metabolism to polyamines production, which reduces the leishmanicidal activity of macrophage. The most remarkable metabolites altered during *Leishmania* infection were proline, glutamic acid, glutamine, L-arginine, ornithine, putrescine, and trypanothione that can be used as a biomarker for infection outcome.

## 4. Materials and Methods

### 4.1. Ethics Statement

The Comissão de Ética no Uso de Animais (CEUA) from the Institute of Bioscience of the University of São Paulo approved the experimental protocol for animal use (approval number CEUA-IB: 233/2014). This study was carried out in strict accordance with the recommendations in the guide and policies for the care and use of laboratory animals of the Brazilian government (Lei Federal 11.794, de 08/10/2008).

### 4.2. Parasite Culture

*L. amazonensis* (MHOM/BR/1973/M2269) wild type promastigotes (La-WT) were maintained in culture at 25 °C in M199 medium (Invitrogen, Grand Island, NY, USA), being supplemented with 10% heat-inactivated fetal bovine serum (Invitrogen), 5 mg/L hemine, 100 µM adenine, 100 U penicillin, 100 µg/mL streptomycin, 40 mM Hepes-NaOH, and 12 mM NaHCO_3_, at pH 6.85, for a week-long culture at a low passage (P1-5). *L. amazonensis* arginase knockout (La-arg^−^) promastigotes were maintained in the same conditions, as previously described, with the medium supplemented with 30 μg/mL hygromycin B, 30 μg/mL puromycin (Sigma, St. Louis, MO, USA), and 50 μM putrescine (Sigma) [29].

### 4.3. In Vitro Macrophage Infections

All of the experiments were performed with 6–8 week old female BALB/c mice obtained from the Animal Center of the Faculty of Medicine of the University of São Paulo and maintained in the Animal Center of the Department of Physiology at the Institute of Bioscience of the University of São Paulo. The bone marrow-derived macrophages (BMDMs) were obtained from the femurs and tibias by flushing with 2 mL of PBS. Subsequently, the cells were collected by centrifugation at 500× *g* for 10 min at 4 °C and then re-suspended in RPMI 1640 medium (LGC Biotecnologia, São Paulo, SP, Brazil), supplemented with penicillin (100 U/mL) (Invitrogen), streptomycin (100 µg/mL) (Invitrogen), 5% heat-inactivated FBS (Invitrogen), and 10% L929 cell supernatant. The cells were submitted to differentiation for 7–8 days at 34 °C in an atmosphere of 5% CO_2_. BMDMs were used after phenotypic analysis by flow cytometry (FACScalibur-Becton Dickinson, San Jose, CA, USA) demonstrated the presence of 95% F4/80- and CD11b-positive cells, which confirmed the macrophage differentiation. 

The BMDMs were seeded into eight-well glass chamber slides (Lab-Teck Chamber Slide; Nunc, Naperville, IL, USA) (2 × 10^5^/well) for infectivity analysis, or into six-well plates (SPL, Lifescience, Pocheon, Korea) (5 × 10^6^/well) for metabolite analysis. After 18 h of incubation at 34 °C in an atmosphere of 5% CO_2_, BMDMs were infected with La-WT or La-arg^−^ promastigotes in the stationary growth phase (MOI 5:1). After 4 h of infection, the non-phagocyted promastigotes were washed with fresh medium and samples were collected for metabolite extraction or fixed for infectivity index determination. The uninfected macrophages were maintained in the same conditions.

The infectivity was microscopically analyzed after cell-fixation with acetone/methanol (1:1, *v:v*, Merck, Darmstadt, Germany) for 20 min at −20 °C, followed by PBS washing and Panoptic-staining (Laborclin, Parana, Brazil). Infectivity was analyzed in phase-contrast microscope (Nikon Eclipse E200, NJ, USA) by counting the number of infected macrophages and amastigotes per macrophage in at least 1000 macrophages/treatment in three independent experiments. The infection index was calculated by multiplying the mean number of amastigotes per macrophage by the rate of macrophage infection. The values were normalized based on the average values for the untreated infected macrophages.

### 4.4. Metabolite Extraction

Pellets with 5 × 10^6^ uninfected or infected macrophages were re-suspended in 350 µL of cold methanol/water (3:1, *v:v*) and 25 mg of glass beads (710–1180 µM, G1152, SigmaAldrich, Germany), followed by four cycles of frost/defrosting in a liquid N_2_/37 °C bath. The cells were disrupted at 50 mHz for 10 min in TissueLyser LT (Qiagen, Germany). The samples were clarified by 15,700× *g* for 10 min at 4 °C centrifugation and the supernatant was collected and evaporated to dryness by SpeedVac SPD121P (Thermo Fisher Scientific, Waltham, MA.) at 35 °C for 2 h. After this, the solid residue was re-suspended in 60 µL 0.1 M formic acid with 0.2 mM of methionine sulfone, homogenized for 15 min in a vortex, and centrifuged at 15,700× *g* for 15 min at 4 °C. The 35 µL of supernatant was transferred to polypropylene vials (Agilent Techno Vials, Waldbronn, Germany) for analysis. Quality-control (QC) samples were prepared by pooling equal volumes of all the samples and they were analyzed along the analytical sequence, to evaluate the stability and performance of the instruments during measurements [27].

### 4.5. CE-MS Metabolic Fingerprinting

The supernatants were analyzed by CE-MS, as previously described [27,63]. The CE-MS analyses were performed in a Capillary Electrophoresis system (7100 Agilent Technologies) coupled to a Time-of-Flight Mass Spectrometer (6224 Agilent Technologies). The coupling was equipped with an electrospray source and an ISO Pump (1200Agilent) to supply the sheath liquid. The separation was performed in a fused-silica capillary (Agilent Technologies) (total length, 100 cm; i.d., 50 μm) in normal polarity with a background electrolyte that was composed of 1.0 mol·L^−1^ of formic acid solution in 10% methanol (*v/v*). New capillaries were pre-conditioned with a flush (950 mbars) of NaOH 1.0 mol·L^−1^ for 30 min, followed by MilliQ water for 30 min and background electrolyte for 30 min. Before each analysis, the capillaries were conditioned with background electrolyte for 5 min. The samples were hydro-dynamically injected at 50 mbar for 50 s and stacking was carried out, applying background electrolyte at 100 mbar for 10 s. The separation voltage that was applied was 30 KV with 25 mbar of internal pressure and the run time was 40 min. The sheath liquid (6 µL·min^−1^) composition was methanol/water (1/1, *v/v*) with two reference masses: 121.0509 purine (C_5_H_4_N_4_) and 922.0098 HP-0921 (C_18_H_18_O_6_N_3_P_3_F_24_), to allow for correction and high mass accuracy in the MS. The MS parameters used were: fragmentor 125 V, Skimmer 65 V, octopole 750 V, nebulizer pressure 10 psi, drying gas temperature at 200 °C, and flow rate 10 mL·min^−1^. The capillary voltage was 3500 V. The data were acquired in positive Dual-ESI mode with a full scan from *m/z* 74 to 1000 at a rate of 1.02 scan/s. MassHunter Workstation version B.06.01 controlled the CE-MS system.

### 4.6. CE-MS Data Processing and Statistical Analysis

Mass Hunter Profinder software processed the raw mass chromatogram was aligned and the peak picking, peak grouping, and retention time correlations (B.08.00, Agilent Technologies, Santa Clara, CA, USA). Data were filtered based on quality while using a quality assurance procedure described previously [64]. This implied retaining features present in at least one of the groups (including QC) at a rate of 70%. For features present in the QC samples, those with CV > 30% were removed in all of the samples. Peak areas from the extracted ion chromatograms were integrated and revised. Multivariate statistical analysis (MVDA) was performed in SIMCA 14.1-1 software (Umetrics, Umea, Sweden) to investigate the pattern of metabolites in *L. amazonensis* WT ((BALB/c)-La-WT) or *L. amazonensis* arginase knockout ((BALB/c)-La-arg^−^)-infected macrophages and uninfected macrophages, as well as in QC controls. The principal component analysis (PCA-X) setting pareto scaling was used to validate the quality of analytical performance, and partial least-squares-discriminant analysis (PLS-DA) and orthogonal partial least-squares-discriminant analysis (OPLS-DA) were carried out for discriminating the variation between the groups, calculating the model’s quality by *R*^2^ and Q^2^. Additionally, we performed an analysis of variance test, ANOVA (comparing three groups), while using in-house algorithms in MatLab (R120151, Mathworks).

## Figures and Tables

**Figure 1 ijms-20-06248-f001:**
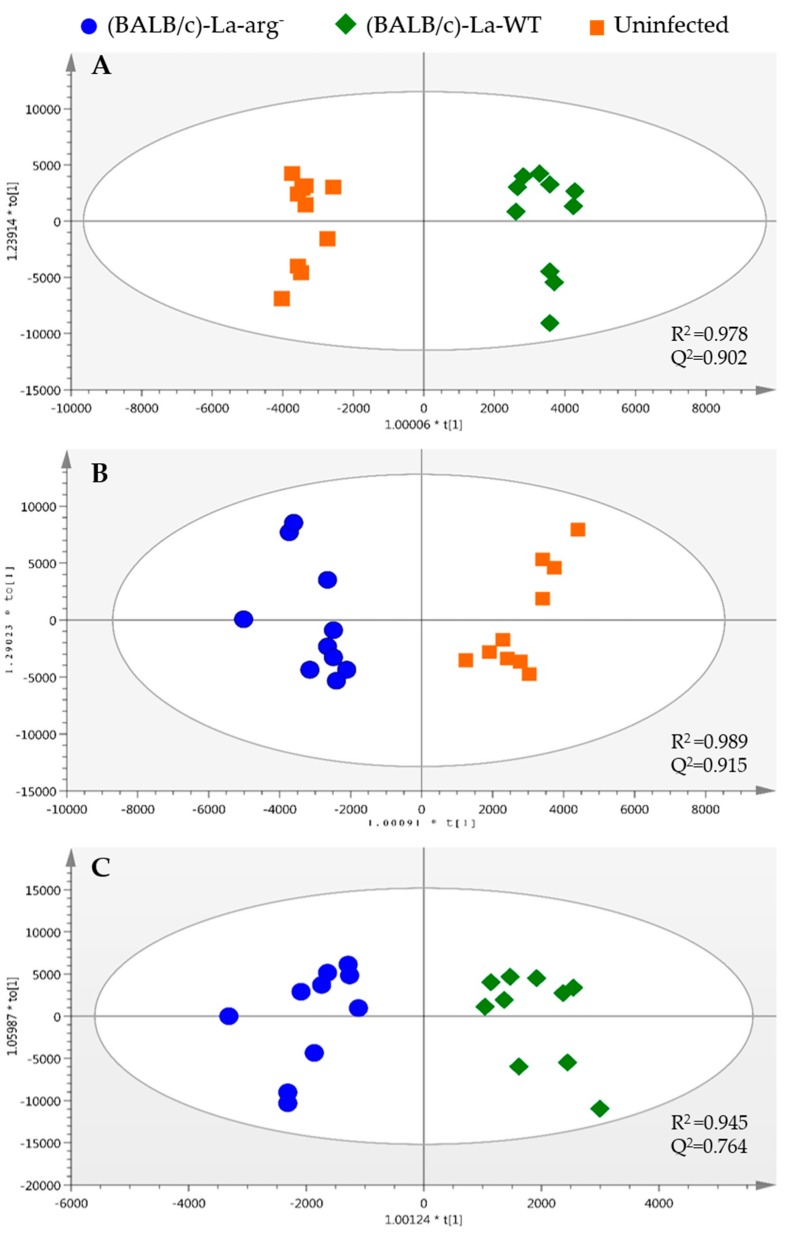
Orthogonal partial least-squares-discriminant analysis (OPLS-DA) analysis of variation between the two groups of *L. amazonensis* infected macrophages. Score plots for the OPLS-DA model were built for three comparisons; (**A**) (BALB/c)-*L. amazonensis* (MHOM/BR/1973/M2269) wild type promastigotes (La-WT) vs. uninfected (*R*^2^ = 0.978 and Q^2^ = 0.902), (**B**) (BALB/c)-La-arg^−^ vs. uninfected (*R*^2^ = 0.989 and Q^2^ = 0.915), and (**C**) (BALB/c)-La-arg^−^ vs. (BALB/c)-La-WT (*R*^2^ = 0.945 and Q^2^ = 0.764). The plot of *L. amazonensis* wild type infected ((BALB/c)-La-WT, green diamond), arginase knockout infected ((BALB/c)-La-arg^−^, blue circle) and uninfected BALB/c (uninfected) macrophages (orange square).

**Figure 2 ijms-20-06248-f002:**
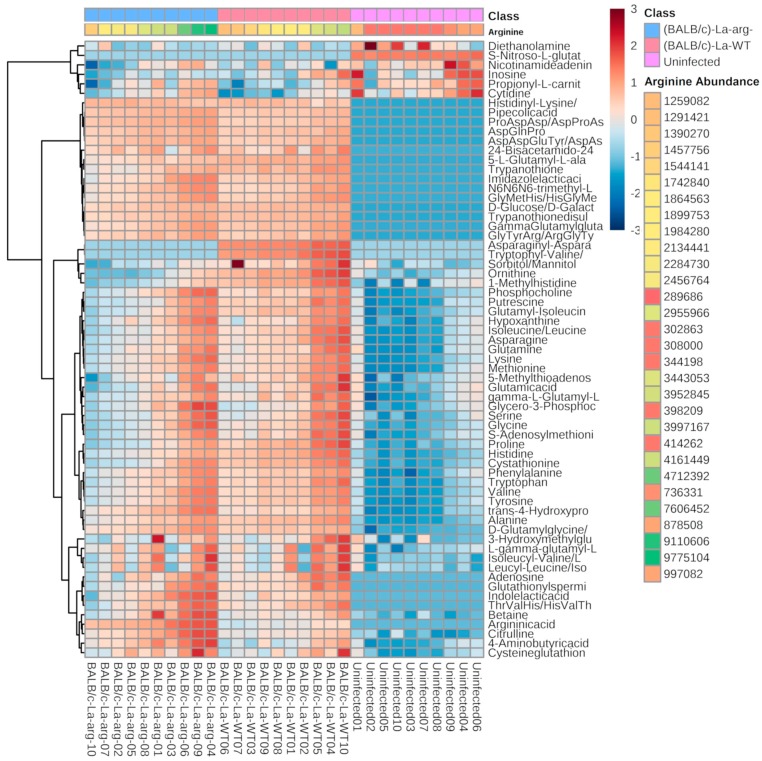
Heatmap analysis of 65 significant metabolites differentially abundant after ANOVA test with a *p* ≤ 0.05 in comparison to uninfected and infected macrophages. Each column represents a sample and each row represents a metabolite. The samples are order by two factors, the first one arranged in three main groups (uninfected, BALB/c-La-arg^−^ and BALB/c-La-WT) and inside each one samples are ordered by the abundance of arginine in each sample. The abundance of arginine has a color from the lowest amount to the highest being red the lowest amount and green the highest. The color code inside the heatmap depicts the relative fold change of each metabolite between groups, red and blue colors express higher or lower abundances, respectively. The parameters that were used for the analysis were Euclidean distance measure and Ward cluster algorithm, using MetaboloAnalyst 4.0 software (http://www.metaboanalyst.ca/faces/Secure/time/Heatmap2View.xhtml).

**Figure 3 ijms-20-06248-f003:**
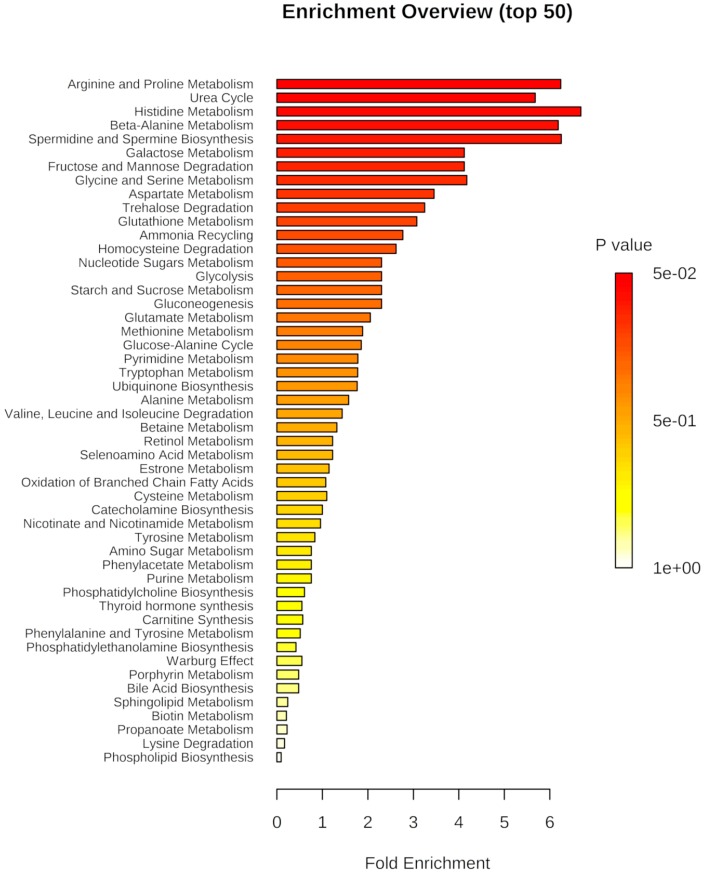
Pathway associated metabolite enrichment in La-arg^−^-infected macrophages versus La-WT-infected macrophages. Enrichment analysis of dysregulated pathways based in metabolite peak areas from *L. amazonensis* wild type ((BALB/c)-La-WT-infected) and arginase knockout ((BALB/c)-La-arg^−^-infected) macrophages using a continuous regression in pathway-associated metabolite sets in MetaboloAnalyst 4.0 software (http://www.metaboanalyst.ca/faces/Secure/time/Heatmap2View.xhtml).

**Figure 4 ijms-20-06248-f004:**
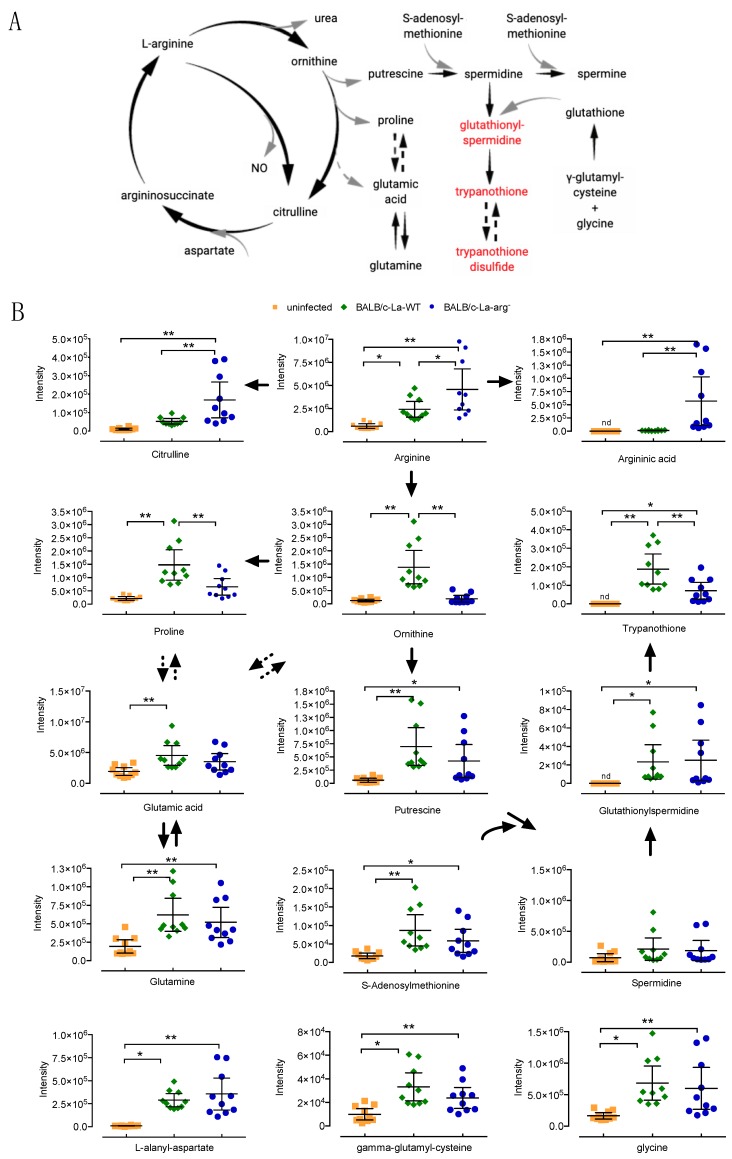
The plot of comparison of metabolites from L-arginine metabolism in *L. amazonensis* infected macrophages. (**A**) Schematic representation of L-arginine and proline metabolism. Metabolites represented in black color are present in macrophages and parasite. Metabolites in red are exclusively found in parasite. The dotted arrows represent the interconversion of proline in glutamic acid or trypanothione in trypanothione disulfide via an intermediate reaction not showed. (**B**) Macrophages (5 × 10^6^) were infected with *L. amazonensis* (MOI 5:1) and collected after 4 h for metabolite quantification. Each point represents the individual values of peak areas (*n* = 10). (*) *p* value < 0.05; (**) *p* value < 0.01. The dotted arrows represent the interconversion of proline in glutamic acid or ornithine in glutamic acid via an intermediate reaction not showed.

**Figure 5 ijms-20-06248-f005:**
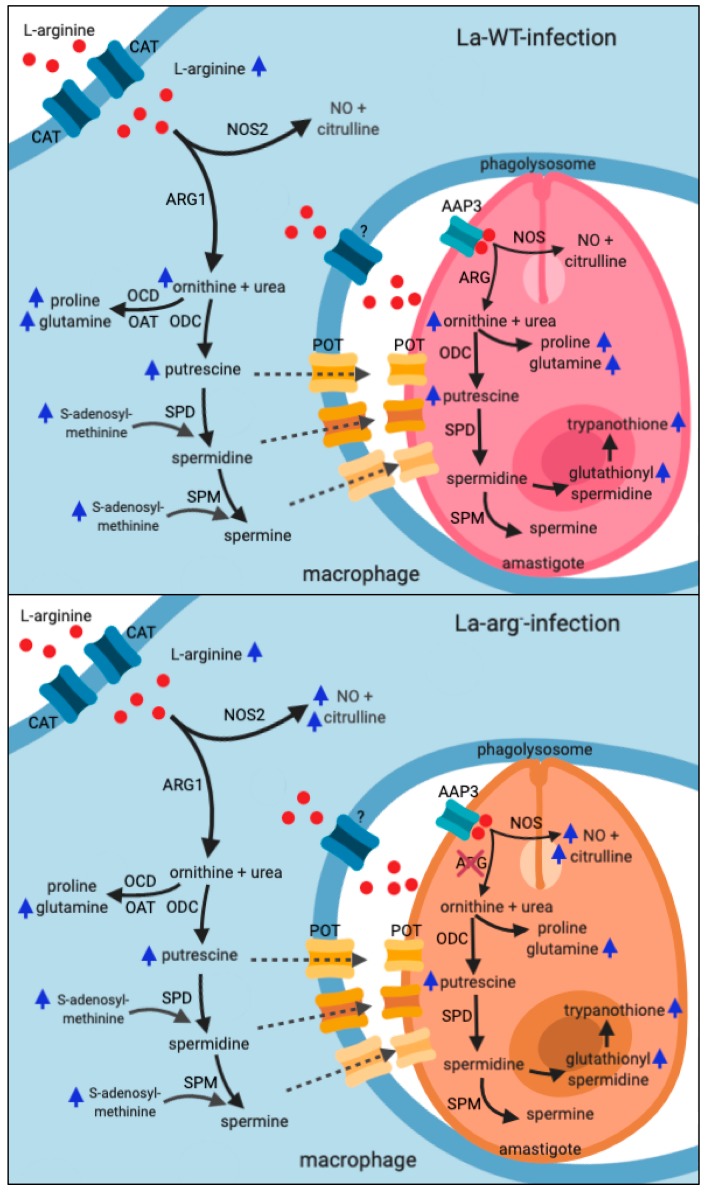
Schematic representation of L-arginine metabolism in *Leishmania*-infected macrophages. The uptake of L-arginine in macrophages occurs via cationic transporter (CAT). In infected-macrophages, depending on L-arginine availability can direct to Nitric Oxide Synthase 2 (NOS2) expression and nitric oxide (NO) and citrulline production, or can direct to arginase 1 (ARG1) expression and ornithine production. Further, ornithine supplies proline, glutamate and putrescine production by ornithine cyclodeaminase (OCD), ornithine aminotransferase (OAT) and ornithine decarboxylase (ODC), respectively. Putrescine and S-adenosyl-methionine are the substrate for spermidine synthase (SpdS) and spermine synthase (SpmS) producing spermidine and spermine, respectively. In infected macrophages, L-arginine transport can occur by polyamine transporter (POT) and other unknown transporters (?). The uptake of L-arginine in *Leishmania* occurs via amino acid permease 3 (AAP3) and depending on L-arginine availability can direct to nitric oxide synthase-like (NOS) expression and NO and citrulline production, or can direct to arginase (ARG) expression and ornithine production. Ornithine supplies proline, glutamate, and putrescine production by OCD, OAT, and ODC, respectively. Putrescine is the substrate for SpdS and SpmS producing spermidine and spermine, respectively. Spermidine is the substrate to produces gluthationyl spermidine and subsequently tripanothine. Blue arrows represent the higher levels of metabolites found in infected-macrophages compared to uninfected.

**Table 1 ijms-20-06248-t001:** Metabolite content differentially identified in the comparison of (BALB/c)-La-WT-infected or (BALB/c)-La-arg^−^-infected macrophages and uninfected macrophages.

						(BALB/c)-La-WT vs. Uninfected				(BALB/c)-La-arg^−^ vs. Uninfected			
**Name**	Mass	RMT	%CV of CQ	*p* ANOVA	*p* FDR	*p* Value	% Change	*P* (Corr)	VIP	*p* Value	% Change	*P* (Corr)	VIP
**Glycine**	75.0328	0.71	3.27	5.5 × 10^−3^	1.5 × 10^−2^	2.6 × 10^−3^	315.1	0.72	<1	9.7 × 10^−3^	263.8	0.56	1.21
**Putrescine**	88.1001	0.41	1.58	3.8 × 10^−3^	1.1 × 10^−2^	9.6 × 10^−4^	1093.3	0.70	<1	4.4 × 10^−2^	624.8	0.52	1.11
**Alanine**	89.0482	0.76	3.64	2.3 × 10^−5^	2.0 × 10^−4^	7.6 × 10^−6^	1964.2	0.85	4.44	4.4 × 10^−4^	1422.9	0.72	3.18
**4-Aminobutyric acid**	103.0646	0.66	3.91	7.5 × 10^−5^	4.6 × 10^−4^	7.9 × 10^−2^	NS	0.61	<1	1.9 × 10^−5^	356.8	0.73	<1
**Serine**	105.0429	0.84	2.02	8.1 × 10^−3^	2.0 × 10^−2^	3.4 × 10^−3^	173.3	0.66	<1	1.6 × 10^−2^	139.0	0.56	1.13
**Diethanolamine**	105.0780	0.65	7.09	1.5 × 10^−2^	3.3 × 10^−2^	1.3 × 10^−2^	−72.0	−0.44	<1	1.0 × 10^−2^	−74.2	0.46	<1
**Proline**	115.0633	0.91	1.72	4.3 × 10^−5^	3.1 × 10^−4^	1.1 × 10^−5^	602.4	0.78	1.40	7.2 × 10^−2^	NS	0.59	1.21
**Valine**	117.0790	0.84	1.37	1.2 × 10^−3^	4.6 × 10^−3^	1.1 × 10^−3^	906.4	0.81	2.29	1.3 × 10^−3^	890.3	0.62	2.62
**Betaine**	117.0792	0.95	6.45	1.9 × 10^−4^	1.0 × 10^−3^	2.3 × 10^−2^	131.0	0.82	<1	4.1 × 10^−5^	264.6	0.74	<1
**Pipecolic acid**	129.0788	0.86	1.26	3.1 × 10^−5^	2.3 × 10^−4^	3.6 × 10^−2^	↑	0.87	1.50	6.9 × 10^−6^	↑	0.73	3.47
***trans*-4-Hydroxyproline**	131.0585	1.01	2.18	2.8 × 10^−4^	1.4 × 10^−3^	8.0 × 10^−3^	725.3	0.81	<1	6.8 × 10^−5^	1190.2	0.69	1.53
**Isoleucine/Leucine**	131.0944	0.87	1.74	2.1 × 10^−3^	6.6 × 10^−3^	5.0 × 10^−4^	467.8	0.70	4.42	3.8 × 10^−2^	257.9	0.56	2.63
**Asparagine**	132.0529	0.88	1.87	1.2 × 10^−3^	4.6 × 10^−3^	2.9 × 10^−4^	493.3	0.68	<1	4.1 × 10^−2^	253.7	0.62	<1
**Ornithine**	132.0896	0.59	4.29	8.4 × 10^−6^	1.2 × 10^−4^	9.8 × 10^−6^	1006.6	0.74	1.42	7.7 × 10^−1^	NS	0.28	<1
**Hypoxanthine**	136.0383	1.01	2.84	7.1 × 10^−3^	1.8 × 10^−2^	5.8 × 10^−3^	391.5	0.66	1.48	5.8 × 10^−3^	391.2	0.59	2.00
**Glutamine**	146.0689	0.91	6.30	2.4 × 10^−3^	7.3 × 10^−3^	8.9 × 10^−4^	221.7	0.69	<1	8.2 × 10^−3^	169.4	0.61	1.03
**Lysine**	146.1053	0.6	2.98	1.5 × 10^−3^	5.0 × 10^−3^	1.0 × 10^−3^	371.2	0.75	1.26	2.1 × 10^−3^	343.1	0.63	1.84
**Glutamic acid**	147.0530	0.92	1.55	9.5 × 10^−3^	2.3 × 10^−2^	2.6 × 10^−3^	135.5	0.65	3.21	5.4 × 10^−2^	NS	0.50	2.33
**Methionine**	149.0511	0.9	2.82	1.9 × 10^−3^	6.1 × 10^−3^	6.6 × 10^−4^	459.2	0.72	<1	8.1 × 10^−3^	340.8	0.61	1.16
**Histidine**	155.0694	0.63	2.33	1.0 × 10^−5^	1.3 × 10^−4^	2.3 × 10^−6^	1552.6	0.80	1.74	2.5 × 10^−2^	615.4	0.68	1.45
**Imidazolelactic acid**	156.0560	0.75	9.11	1.5 × 10^−2^	3.3 × 10^−2^	8.7 × 10^−2^	NS	NS	NS	4.1 × 10^−3^	↑	0.53	<1
**3-Hydroxymethylglutaric acid/2-Hydroxyadipic acid/2-Ethylmalate**	162.0524	1.81	6.41	3.3 × 10^−2^	6.9 × 10^−2^	1.5 × 10^−2^	72.8	0.61	<1	3.8 × 10^−2^	61.4	0.60	<1
**Phenylalanine**	165.0803	0.93	1.84	4.1 × 10^−4^	1.7 × 10^−3^	2.9 × 10^−4^	538.5	0.82	1.05	8.2 × 10^−4^	487.2	0.65	1.71
**1-Methylhistidine**	169.0863	0.65	4.07	1.2 × 10^−6^	2.5 × 10^−5^	1.0 × 10^−6^	549.1	0.79	<1	4.7 × 10^−1^	NS	0.34	<1
**Arginine**	174.1116	0.62	2.43	3.9 × 10^−4^	1.7 × 10^−3^	4.5 × 10^−2^	306.5	0.74	2.04	8.8 × 10^−5^	670.4	0.68	3.62
**Argininic acid**	175.0966	0.78	3.86	2.3 × 10^−3^	7.2 × 10^−3^	9.5 × 10^−1^	NS	NS	NS	2.0 × 10^−3^	↑	0.55	1.38
**Citrulline**	175.0975	0.93	2.33	3.9 × 10^−4^	1.7 × 10^−3^	2.5 × 10^−1^	NS	0.83	<1	1.3 × 10^−4^	1497.9	0.64	<1
**α-d-Glucose/d-Galactose/d-Fructose/d-(+)-Mannose/myo-Inositol**	180.0608	1.82	13.22	3.5 × 10^−7^	8.3 × 10^−6^	1.1 × 10^−7^	↑	0.90	<1	2.6 × 10^−5^	↑	0.78	1.07
**Tyrosine**	181.0738	0.95	1.81	3.1 × 10^−4^	1.5 × 10^−3^	2.2 × 10^−4^	543.7	0.84	<1	6.6 × 10^−4^	490.4	0.65	1.39
**Sorbitol/Mannitol**	182.0802	1.81	7.76	1.0 × 10^−4^	5.9 × 10^−4^	8.1 × 10^−5^	82.3	0.72	<1	6.8 × 10^−1^	NS	0.39	<1
**Phosphocholine**	183.0660	1.69	2.86	1.4 × 10^−3^	4.9 × 10^−3^	1.6 × 10^−3^	409.2	0.80	1.36	1.3 × 10^−3^	418.7	0.62	2.06
***N*_6_, *N*_6_, *N*_6_-trimethyl-L-lysine**	188.1511	0.62	9.39	1.3 × 10^−3^	4.6 × 10^−3^	4.2 × 10^−3^	↑	0.73	<1	5.3 × 10^−4^	↑	0.67	<1
**γ-D-Glutamylglycine/L-beta-aspartyl-L-alanine/Alanyl-Aspartate/Aspartyl-Alanine**	204.0744	1.01	2.53	4.9 × 10^−5^	3.3 × 10^−4^	3.4 × 10^−4^	2788.0	0.91	<1	2.3 × 10^−5^	3469.4	0.72	1.07
**Tryptophan**	204.0895	0.93	2.19	1.3 × 10^−4^	6.9 × 10^−4^	4.5 × 10^−5^	431.3	0.81	<1	1.2 × 10^−3^	320.6	0.68	<1
**Indolelactic acid**	205.0735	1.85	6.11	7.6 × 10^−3^	1.9 × 10^−2^	3.1 × 10^−1^	NS	NS	NS	2.4 × 10^−3^	↑	0.54	<1
**Propionyl-L-carnitine**	217.1340	0.79	21.78	1.1 × 10^−2^	2.6 × 10^−2^	3.5 × 10^−3^	−46.7	−0.60	<1	3.9 × 10^−2^	−31.7	−0.42	<1
**5-L-Glutamyl-L-alanine/Hydroxyprolyl-Serine**	218.0909	1.03	4.53	1.4 × 10^−10^	7.9 × 10^−9^	3.4 × 10^−11^	↑	0.93	<1	9.8 × 10^−4^	↑	0.88	<1
**Cystathionine**	222.0682	0.85	1.52	1.2 × 10^−4^	6.6 × 10^−4^	2.6 × 10^−5^	1112.1	0.84	<1	9.5 × 10^−3^	613.3	0.57	<1
**Isoleucyl-Valine/Leucyl-Valine/Valyl-Isoleucine/Valyl-Leucine**	230.1618	0.89	5.45	1.1 × 10^−2^	2.5 × 10^−2^	1.0 × 10^−2^	150.3	0.67	<1	7.0 × 10^−3^	159.3	0.63	<1
**Cytidine**	243.0889	0.82	5.47	5.8 × 10^−3^	1.5 × 10^−2^	1.6 × 10^−3^	−67.6	−0.59	<1	3.8 × 10^−2^	−42.0	−0.38	<1
**Leucyl-Leucine/Isoleucyl-Leucine/Leucyl-Isoleucine**	244.1798	0.9	6.33	3.0 × 10^−2^	6.3 × 10^−2^	1.4 × 10^−2^	192.3	0.61	<1	3.5 × 10^−2^	162.7	0.57	<1
**Asparaginyl-Asparagine/*N*_2_-Oxalylarginine**	246.0966	1.26	3.24	2.9 × 10^−7^	8.1 × 10^−6^	6.7 × 10^−7^	↑	0.79	<1	1.0 × 10^0^	NS	NS	NS
**2,4Bis(acetamido)2,4,6trideoxy-betaL-altropyranose/L-beta-aspartyl-L-leucine/L-gamma-glutamyl-L-valine/Aspartyl-Isoleucine/aspartyl-Leucine/Isoleucyl-Aspartate/LeucylAspartate/Glu-Val**	246.1227	0.91	10.84	2.6 × 10^−3^	7.8 × 10^−3^	1.3 × 10^−3^	↑	0.72	<1	4.9 × 10^−3^	↑	0.72	<1
**gamma-L-Glutamyl-L-cysteine**	250.0628	1.07	15.49	1.2 × 10^−3^	4.6 × 10^−3^	3.1 × 10^−4^	233.1	0.71	<1	2.1 × 10^−2^	138.7	0.58	<1
**Glycero-3-Phosphocholine**	257.1028	1.78	4.73	1.5 × 10^−2^	3.3 × 10^−2^	3.6 × 10^−2^	136.4	0.61	<1	5.3 × 10^−3^	188.1	0.57	1.69
**L-gamma-glutamyl-L-isoleucine**	260.1390	0.91	7.49	4.7 × 10^−2^	9.1 × 10^−2^	1.9 × 10^−2^	427.3	0.57	<1	6.8 × 10^−2^	NS	0.55	<1
**Adenosine**	267.0975	0.84	5.60	3.4 × 10^−2^	7.0 × 10^−2^	1.2 × 10^−2^	↑	0.56	<1	7.3 × 10^−2^	NS	0.54	<1
**Inosine**	268.0799	1.65	2.07	2.5 × 10^−2^	5.4 × 10^−2^	7.3 × 10^−2^	NS	−0.36	<1	8.0 × 10^−3^	−78.4	−0.53	<1
**Gamma Glutamylglutamic acid**	276.0941	1.1	7.21	9.8 × 10^−5^	5.8 × 10^−4^	6.0 × 10^−5^	↑	0.82	<1	3.5 × 10^−4^	↑	0.71	<1
**Histidinyl-Lysine/Lysyl-Histidine**	283.1573	0.83	28.48	1.3 × 10^−11^	1.1 × 10^−9^	1.0 × 10^−9^	↑	0.90	<1	7.5 × 10^−12^	↑	0.95	<1
**5’-Methylthioadenosine**	297.0882	0.86	25.00	4.1 × 10^−3^	1.2 × 10^−2^	1.0 × 10^−3^	262.7	0.67	<1	9.4 × 10^−2^	NS	0.50	<1
**Tryptophyl-Valine/Valyl-Tryptophan**	303.1548	0.75	6.01	1.4 × 10^−5^	1.6 × 10^−4^	2.3 × 10^−5^	↑	0.71	<1	1.0 × 10^0^	NS	NS	NS
**S-Nitroso-L-glutathione**	336.0745	1.12	ND	5.0 × 10^−18^	8.3 × 10^−16^	3.0 × 10^−17^	↓	−0.96	<1	3.0 × 10^−17^	↓	−0.96	<1
**Gly Met His/His Gly Met/Met His Gly/Met Gly His/Gly His Met/His Met Gly**	343.1361	0.84	11.05	4.7 × 10^−3^	1.3 × 10^−2^	3.4 × 10^−3^	↑	0.69	<1	4.9 × 10^−3^	↑	0.61	<1
**Pro Asp Asp/Asp Pro Asp/Asp Asp Pro**	345.1186	1.8	2.76	3.0 × 10^−6^	5.0 × 10^−5^	1.6 × 10^−2^	↑	0.85	<1	6.4 × 10^−7^	↑	0.79	1.32
**Thr Val His/His Val Thr/Leu Ser His/His Thr Val/Val Thr His/His Ile Ser/Ser His Ile/His Leu Ser/Ile Ser His/Val His Thr/His Ser Leu/His Ser Ile/Ser Ile His/Ser Leu His/Ser His Leu/Ile His Ser/Leu His Ser/Thr His Val**	355.1791	0.84	23.12	3.6 × 10^−2^	7.2 × 10^−2^	9.4 × 10^−2^	NS	0.59	<1	1.1 × 10^−2^	↑	0.51	<1
**Asp Gln Pro**	358.1424	0.83	11.91	1.5 × 10^−7^	5.1 × 10^−6^	2.7 × 10^−6^	↑	0.88	<1	8.9 × 10^−8^	↑	0.86	<1
**Gly Tyr Arg/Arg Gly Tyr/Tyr Gly Arg/Gly Arg Tyr/Arg Tyr Gly/Tyr Arg Gly**	394.1954	0.99	5.96	1.9 x 10^−5^	1.9 × 10^−4^	2.8 × 10^−5^	↑	0.84	<1	3.3 × 10^−5^	↑	0.80	<1
**S-Adenosylmethionine**	398.1381	0.62	2.50	4.7 × 10^−3^	1.3 × 10^−2^	1.3 × 10^−3^	389.4	0.67	<1	4.2 × 10^−2^	230.3	0.55	<1
**Cysteineglutathione disulfide**	426.0907	0.98	27.11	2.4 × 10^−2^	5.1 × 10^−2^	5.0 × 10^−2^	253.3	0.54	<1	8.3 × 10^−3^	351.3	0.60	<1
**Glutathionylspermidine**	434.2371	0.64	15.79	4.1 × 10^−2^	8.2 × 10^−2^	3.4 × 10^−2^	↑	0.58	<1	2.3 × 10^−2^	↑	0.52	<1
**Asp Asp Glu Tyr/Asp Asp Tyr Glu/Asp Glu Asp Tyr/Asp Glu Tyr Asp/Asp Tyr Asp Glu/Asp Tyr Glu Asp/Glu Asp Asp Tyr/Glu Asp Tyr Asp/Glu Tyr Asp Asp/Tyr Asp Asp Glu/Tyr Asp Glu Asp/Tyr Glu Asp Asp**	540.1670	0.83	3.91	7.7 × 10^−6^	1.2 × 10^−4^	1.3 × 10^−4^	↑	0.82	<1	3.0 × 10^−6^	↑	0.83	<1
**Nicotinamide adenine dinucleotide (NAD)**	663.1087	1.81	4.11	1.6 × 10^−2^	3.5 × 10^−2^	1.0 × 10^−2^	−51.8	−0.44	<1	1.4 × 10^−2^	−49.4	−0.42	<1
**Trypanothione disulfide**	721.2894	0.81	13.64	2.2 × 10^−4^	1.1 × 10^−3^	6.4 × 10^−5^	↑	0.81	<1	3.2 × 10^−3^	↑	0.66	1.34
**Trypanothione**	723.3058	0.83	26.04	2.6 × 10^−5^	2.2 × 10^−4^	6.1 × 10^−6^	↑	0.78	<1	4.4 × 10^−2^	↑	0.63	<1

↑, represents the presence of the metabolite in La-WT-infected macrophages or La-arg^−^-infected macrophages and the absence in the uninfected macrophages; ↓, represents the presence of the metabolite in uninfected macrophages and the absence in infected macrophages. % change, represents the increase (positive values) or decrease (negative values) in metabolite abundance in the infected groups, this was calculated as follows ((Average of infected macrophage) − (Average of uninfected)/(Average of uninfected)) × 100. VIP (Variable Importance in Projection) represents the weight of that metabolite in the separation in the OPLS-DA model, and values higher than 1 reflect strong differences of that metabolite between the groups. These values were calculated, by default, from all extracted compound data using SIMCA 14.1-1 software. NS, non-significant.

**Table 2 ijms-20-06248-t002:** Metabolite content differentially identified in the comparison of (BALB/c)-La-arg^−^-infected macrophages versus (BALB/c)-La-WT-infected.

			(BALB/c)-La-arg^−^ vs. (BALB/c)-La-WT						
**Name**	MASS	RMT	% CV of QC	*p* ANOVA	*p* FDR	*p* Value	% Change	*p* (Corr)	VIP
4-Aminobutyric acid	103.0646	0.66	3.91	7.5 × 10^−5^	4.6 × 10^−4^	2.4 × 10^−3^	102.17	0.56	<1
Proline	115.0633	0.91	1.72	4.3 × 10^−5^	3.1 × 10^−4^	1.6 × 10^−3^	−55.90	−0.59	1.74
Betaine	117.0792	0.95	6.45	1.9×10^−4^	1.0 × 10^−3^	2.0 × 10^−2^	57.84	0.50	<1
Pipecolic acid	129.0788	0.86	1.26	3.1 × 10^−5^	2.3 × 10^−4^	2.4 × 10^−3^	152.06	0.56	4.90
Ornithine	132.0896	0.59	4.29	8.4 × 10^−6^	1.2 × 10^−4^	2.2 × 10^−5^	−86.03	−0.75	2.44
Histidine	155.0694	0.63	2.33	1.0 × 10^−5^	1.3 × 10^−4^	1.2 × 10^−3^	−56.71	−0.59	2.03
1-Methylhistidine	169.0863	0.65	4.07	1.2 × 10^−6^	2.5 × 10^−5^	7.0 × 10^−6^	−74.72	−0.76	<1
Arginine	174.1116	0.62	2.43	3.9 × 10^−4^	1.7 × 10^−3^	1.9 × 10^−2^	89.54	0.44	5.13
Argininic acid	175.0966	0.78	3.86	2.3 × 10^−3^	7.2 × 10^−3^	2.4 × 10^−3^	4881.40	0.56	1.24
Citrulline	175.0975	0.93	2.33	3.9 × 10^−4^	1.7 × 10^−3^	3.0 × 10^−3^	220.22	0.55	<1
α-D-Glucose/D-Galactose/D-Fructose/D-(+)-Mannose/myo-Inositol	180.0608	1.82	13.22	3.5 × 10^−7^	8.3 × 10^−6^	4.5 × 10^−2^	−29.34	−0.35	<1
Sorbitol/Mannitol	182.0802	1.81	7.76	1.0 × 10^−4^	5.9 × 10^−4^	2.5 × 10^−4^	−41.14	−0.61	<1
Indolelactic acid	205.0735	1.85	6.11	7.6 × 10^−3^	1.9 × 10^−2^	2.9 × 10^−2^	221.22	0.43	<1
5-L-Glutamyl-L-alanine/Hydroxyprolyl-Serine	218.0909	1.03	4.53	1.4 × 10^−10^	7.9 × 10^−9^	1.6 × 10^−7^	−65.39	−0.79	<1
Cystathionine	222.0682	0.85	1.52	1.2 × 10^−4^	6.6 × 10^−4^	3.1 × 10^−2^	−41.16	−0.38	<1
Asparaginyl-Asparagine/*N*_2_-Oxalylarginine	246.0966	1.26	3.24	2.9 × 10^−7^	8.1 × 10^−6^	6.7 × 10^−7^	↓	−0.81	<1
Histidinyl-Lysine/Lysyl-Histidine	283.1573	0.83	28.48	1.3 × 10^−11^	1.1 × 10^−9^	2.8 × 10^−2^	25.52	0.37	<1
Tryptophyl-Valine/Valyl-Tryptophan	303.1548	0.75	6.01	1.4 × 10^−5^	1.6 × 10^−4^	2.3 × 10^−5^	↓	−0.76	<1
Pro Asp Asp/Asp Pro Asp/Asp Asp Pro	345.1186	1.8	2.76	3.0 × 10^−6^	5.0 × 10^−5^	5.8 × 10^−4^	152.34	0.62	<1
Trypanothione	723.3058	0.83	26.04	2.6 × 10^−5^	2.2 × 10^−4^	1.7 × 10^−3^	−62.23	−0.57	<1

↓, represents the presence of the metabolite in La-arg^−^-infected macrophages compared to La-WT-infected macrophages. % change, represents the increase (positive values) or decrease (negative values) in metabolite abundance in the infected groups, this was calculated, as follows ((Average of BALB/c-La-arg^−^) − (Average of BALB/c-La-WT)/(Average of BALB/c-La-WT)) × 100. VIP (Variable Importance in Projection) represents the weight of that metabolite in the separation in the OPLS-DA model, and values higher than 1 reflect strong differences of that metabolite between the groups. These values were calculated, by default, from all extracted compound data using SIMCA 14.1-1 software.

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
