# Peer review of "Metabolomic Profile of BALB/c Macrophages Infected with Leishmania amazonensis: Deciphering L-Arginine Metabolism"

_ijms, 2019, doi:10.3390/ijms20246248_

Round 1

Reviewer 1 Report

Here, the authors utilize metabolomics approach to test small molecule levels in mouse macrophages with treatments: Untreated (UT), Leishmania amazonensis WT (La-WT) and L. amazonensis Arginase k.o. (La-Arg-). They observe several dozen molecules with differential levels between the three treatments; they focus subsequently on differences in Arginine metabolism and focus on differences between La-WT and UT. As expected in such a treatment, there are extensive differences between the treatments’ macrophage metabolites, including in Arginine metabolism. 

Due to previous work cited heavily by the authors, who have a pretty extensive experience with Leishmania infection and interaction with macrophages, the work in this paper has limited value added to the field: In the introduction, the authors cite references 18-20 (Lines 56-60in particular) as evidence that La-WT significantly affects arginine metabolism, and the use of La-Arg- previously to demonstrate that disruption of Arg metabolism significantly reduces infection by La. It’s unclear, therefore, what is really added in this paper, other than specific levels of individual metabolites, particularly in the Arg metabolism pathway. For example, lines 132-134 in which the authors state that “...only 20 metabolites appeared regulated in comparison of La-Arg- infected vs La-WT infected....absence of parasite arginase activity impacted on the metabolite concentrations” - this appears known from reference 18, and certainly can be implied. Similarly, lines 140-145 in which the authors state that arginine and proline metabolism are affected by La-WT infection, which they’ve already noted is known.

Figure 4 was the most informative of the paper, and shows a further problem that I did not find addressed sufficiently: there are significant differences in Arg metabolites between the three treatments, but only two (ornithine and proline) significantly differ between the two La treatments. While the authors provide a nod to this (Lines 237-42), it is to me insufficient to address the lack of difference in downstream metabolite levels when the authors appear to be set on using the Arg metabolites as a buttress for their case that Arg metabolism is necessary to La-WT infection (and presumably, the alterations in Arg metabolites is why La-Arg- are less infectious). Analysis of enzymes involved in the synthesis of these downstream components would add to this work.

Given this is a general science journal with a special focus on arginine metabolism, the authors would do well to include a figure describing the parasite’s Arg metabolism and highlighting their hypothesis here. As it stands, the introduction is difficult to follow, particularly in light of frequent lack of clarity in language usage. 

Most problematic in language was the frequent use of the term “regulated” (eg, Line 115, 117): this term connotes a direct interaction and effect, while the authors’ data show what is in many cases an effect, which can come about through numerous ways (eg, direct parasite production and manipulation, or indirectly via host manipulation or even more by host response). 

The authors would do well to describe in the results their methods in a little more detail. FOr example, they use the “infectivity index”, but do not define outside of the methods which of course come after the results. Briefly describing the methods as they go into the results generated from them would bolster their interpretation in some cases, and certainly the readers’ understanding. 

Line 96: R2 should be either percent or decimal - I don’t think the authors meant “....good quality, R2 > 0.945%....”

It’s unclear in Figure 2 what the individual columns represent - presumably individual samples within the three treatments, but this should be explicitly described. Additionally, the color code (red = 3, blue = -3) should be exploit: the ratio/fold change is relative to what?

Figure 3: I don’t understand how the authors can show a “Fold enrichment” was shown in 2D for 3 comparisons (UT, La-WT, La-Arg-) as implied by lines 148-150, unless it’s really just UT vs La-WT as implied by Lines 147-148. In which case, why is La-Arg- noted in line 149?

Line 177-78: the authors state that macrophages are specialized to deal with Leishmania. This is inaccurate: macrophages are professional immune cells that deal with all sorts of parasites and foreign bodies, some of which might be Leishmania spp.

Line 198-99: unclear what the authors mean by “leading to interference in host-cell gene expression”. 

Caption for figure S2: “...using pareto setting in non-normalized...” appears incomplete

Author Response

Author's Reply to the Review Report (Reviewer 1)

Here, the authors utilize metabolomics approach to test small molecule levels in mouse macrophages with treatments: Untreated (UT), Leishmania amazonensis WT (La-WT) and L. amazonensis Arginase k.o. (La-Arg-). They observe several dozen molecules with differential levels between the three treatments; they focus subsequently on differences in Arginine metabolism and focus on differences between La-WT and UT. As expected in such a treatment, there are extensive differences between the treatments’ macrophage metabolites, including in Arginine metabolism. 

Due to previous work cited heavily by the authors, who have a pretty extensive experience with Leishmania infection and interaction with macrophages, the work in this paper has limited value added to the field:

In the introduction, the authors cite references 18-20 (Lines 56-60in particular) as evidence that La-WT significantly affects arginine metabolism, and the use of La-Arg- previously to demonstrate that disruption of Arg metabolism significantly reduces infection by La. It’s unclear, therefore, what is really added in this paper, other than specific levels of individual metabolites, particularly in the Arg metabolism pathway.

Response: We appreciate the comments made by the reviewer #1 and we tried to clarify all the points raised, to make the manuscript clear.

In the introduction we pointed previous work that focused on the L-arginine metabolism of the parasite. In the present manuscript, we described the metabolomic profile of macrophages infected with L. amazonensis, considering that the metabolism of host and parasite intercross. Considering that, we also studied the impact of parasite arginase into macrophage metabolism. So, in our point of view, this manuscript describes a new and important contribution to Leishmania field.

For example, lines 132-134 in which the authors state that “...only 20 metabolites appeared regulated in comparison of La-Arg- infected vs La-WT infected....absence of parasite arginase activity impacted on the metabolite concentrations” - this appears known from reference 18, and certainly can be implied. Similarly, lines 140-145 in which the authors state that arginine and proline metabolism are affected by La-WT infection, which they’ve already noted is known.

Response: We reinforce that the present manuscript refers the description of metabolites from macrophages infected with L. amazonensis WT(La-WT) or infected with L. amazonensis arginase knockout, compared to uninfected macrophages. It is a new aspect of analysis, different from described in reference 18 that encompass only the parasite metabolomics profile.

Figure 4 was the most informative of the paper, and shows a further problem that I did not find addressed sufficiently: there are significant differences in Arg metabolites between the three treatments, but only two (ornithine and proline) significantly differ between the two La treatments. While the authors provide a nod to this (Lines 237-42), it is to me insufficient to address the lack of difference in downstream metabolite levels when the authors appear to be set on using the Arg metabolites as a buttress for their case that Arg metabolism is necessary to La-WT infection (and presumably, the alterations in Arg metabolites is why La-Arg- are less infectious). Analysis of enzymes involved in the synthesis of these downstream components would add to this work.

Response: We change the discussion (lines 247-260) pointing the possible mediator role that the reduction of ornithine and proline levels can cause in downstream metabolites, indicating  the substrates, enzymes and product informations.

Given this is a general science journal with a special focus on arginine metabolism, the authors would do well to include a figure describing the parasite’s Arg metabolism and highlighting their hypothesis here. As it stands, the introduction is difficult to follow, particularly in light of frequent lack of clarity in language usage. 

Response: We changed the introduction section on trial to clarify the L-arginine metabolism in the context of parasite, as well as in the host-cells. Additionally, we designed the Figure 5 to summarize the L-arginine metabolism based on our results.

Most problematic in language was the frequent use of the term “regulated” (eg, Line 115, 117): this term connotes a direct interaction and effect, while the authors’ data show what is in many cases an effect, which can come about through numerous ways (eg, direct parasite production and manipulation, or indirectly via host manipulation or even more by host response). 

Response: As requested by the Reviewer #1, we changed that phrases to clarify the context of change in the metabolites abundance during infection of macrophages: lines

“From those, 58 metabolites appeared modulated in La-WT-infected compared to uninfected macrophages, indicating that L. amazonensis infection impacted on metabolite abundance, which can be mediate by its own metabolism inside macrophages and/or manipulate the host metabolism. Indeed, 55 metabolites appeared modified in La-arg--infected compared to uninfected macrophages, as consequence of parasite metabolism into host cell and macrophage activation of microbicide metabolism (Table 1).”

The authors would do well to describe in the results their methods in a little more detail. For example, they use the “infectivity index”, but do not define outside of the methods which of course come after the results. Briefly describing the methods as they go into the results generated from them would bolster their interpretation in some cases, and certainly the readers’ understanding. 

Response:

Line 96: R2 should be either percent or decimal - I don’t think the authors meant “....good quality, R2 > 0.945%....”

Response: We thank for the comment. The reviewer is right, this was a typo mistake, it has been corrected to R2 ≥ 0.945 and Q2 ≥ 0.764.

The R2 provided by the software SIMCA P+, which explains variance, ranges from 0 to 1 and gives information about goodness of fit. The value of the R2 increases with increased number of principal components because the R2 that we have provided is the R2X(cum) which corresponds to the cumulative R2X up to the specified component. The R2X is the fraction of X variation modeled in a component. It is stablished that R2 should be higher than 0.7 together with a Q2 > 0.4 for metabolomics data to ensure the model is powerful for diagnostics (1). Therefore, we can assume that the model we obtained in this study was good based on the R2 provided (higher than 0.7). 

Reference: (1) Godzien, J.; Ciborowski, M.; Angulo, S.; Barbas, C. Electrophoresis 2013, 34, 2812-2826.

It’s unclear in Figure 2 what the individual columns represent - presumably individual samples within the three treatments, but this should be explicitly described. Additionally, the color code (red = 3, blue = -3) should be exploit: the ratio/fold change is relative to what?

Response: We fully agree with the reviewer. The heatmap figure was made with the 65 metabolites from the ANOVA analysis with a p value ≤ 0.05. Each column represents a sample and each row represents a metabolite. The samples are order by two factors, the first one arranged in three main groups (uninfected, BALB/c-La-arg- and BALB/c-La-WT) and inside each one samples are ordered by the abundance of arginine in each sample. The abundance of arginine has a color from the lowest amount to the highest being red the lowest amount and green the highest. In the case of metabolites, which are the rows, these are distributed according to the hierarchical algorithm. The color code depicts the fold change of each metabolite as a relative value between groups. Although Metaboanalyst, the webpage used to obtain the graphic, does not explain how it is obtained exactly, we supposed the fold change is relative to the median of the three groups, leading in the colors red and blue to express higher or lower abundances, respectively. Following the reviewer comments, the figure and its capture have been changed in order to clarify the figure 2.

Figure 3: I don’t understand how the authors can show a “Fold enrichment” was shown in 2D for 3 comparisons (UT, La-WT, La-Arg-) as implied by lines 148-150, unless it’s really just UT vs La-WT as implied by Lines 147-148. In which case, why is La-Arg- noted in line 149?

Response: We change the Figure 3 and Suplemmentary Table 1 with Enrichment Analysis of dysregulated pathways based in metabolites peak areas from L. amazonensis wild type ((BALB/c)-La-WT-infected) and arginase knockout ((BALB/c)-La-arg--infected) macrophages using a continuous regression in pathway-associated metabolite sets in MetaboloAnalyst 4.0 software (http://www.metaboanalyst.ca/faces/Secure/time/Heatmap2View.xhtml).

Also, We included the Enrichment Analysis of dysregulated pathways based in metabolites peak areas from La-WT-infected compared with uninfected macrophages (Figure S3, Table S2); and La-arg-infected macrophages and uninfected macrophages (Figure S4, Table S3).

The Fold Enrichment FDR is calculated by dividing the observed Q statistic by the expected Q statistic ("Statistic" / "Expected" columns of the results table).

Line 177-78: the authors state that macrophages are specialized to deal with Leishmania. This is inaccurate: macrophages are professional immune cells that deal with all sorts of parasites and foreign bodies, some of which might be Leishmania spp.

Response: The reviewer #1 is right. We changed that sentence, describing how macrophages are specialized cells that can recognize and phagocytize several microrganisms, including Leishmania (see the revised version of the manuscript).  

Line 198-99: unclear what the authors mean by “leading to interference in host-cell gene expression”. 

Response: We changed that sentence to clarify, lines 207-208: “L-arginine availability and arginase activity play an important role in gene expression and metabolite from L-arginine and polyamines metabolism in Leishmania promastigote and amastigote forms and also impacts in parasite survival and growth, [16,18,19].”

Caption for figure S2: “...using pareto setting in non-normalized...” appears incomplete

Response: We thank the reviewer for this comment. The caption was corrected (see the revised version of the manuscript).

Reviewer 2 Report

The manuscript is interesting and well-structured in general. The Authors conducted experiments to reveal the metabolomic profile of BALB/c macrophages infected with Leishmania amazonensis. In my opinion the manuscript could be interesting for a reasonable number of scientists since Leishmaniases are neglected tropical diseases that are endemic around the world. However, before the manuscript could be published the Authors should undertake revision that in my opinion would improve their study:

1. The potential use of the obtained data should be more highlighted - it concers Abstract as well as the the Discussion (Conclusion) sections.  

In my opinion after this correction the manuscript merit publication.

Author Response

Author's Reply to the Review Report (Reviewer 2)

The manuscript is interesting and well-structured in general. The Authors conducted experiments to reveal the metabolomic profile of BALB/c macrophages infected with Leishmania amazonensis. In my opinion the manuscript could be interesting for a reasonable number of scientists since Leishmaniases are neglected tropical diseases that are endemic around the world. However, before the manuscript could be published the Authors should undertake revision that in my opinion would improve their study:

The potential use of the obtained data should be more highlighted - it concers Abstract as well as the the Discussion (Conclusion) sections.   

Response: We appreciate the comments made by the reviewer #2 and we tried to improve the abstract and the discussion sections on trial to highlighted our main findings (see the revised version of the manuscript).

Round 2

Reviewer 1 Report

I reviewed a previous submission of this manuscript. This resubmission is highly improved, with the authors taking care to address virtually all concerns I had regarding the original submission. They have introduced numerous clarifications as well as been more precise in their language. A number of issues still remain but they are minor and should be addressable by the authors. All in all, much improved.

Specifics:

p5, lines 133-140: authors state that statistical analyses identified 65 metabolites undergoing changes; I count 67 metabolites in Table 1 (but conversely, there are 65 in Fig 2 heatmap of the data from Table 2?!). Table 1: Clarify what column "% change" and VIP mean - former should note what the comparison is made between (specifically, is the infected or uninfected the numerator or denominator?), latter should be spelled out/defined in Table description.  Meaning of upward arrows in "% change" column should be defined in table description. Table 2: what is the reference of change to: is it the Arg- (nominator) vs WT (denominator)? This should be clarified. I don't see a Non-significant p value, so "NS, non-significant" not required again, what is down arrow mean? Missing two column headers to the right of "VIP" column p11, Line 171: trehalose appears to be misspelled Line 177-185: this section is unclear on whether the comparison is Arg- vs WT, or Arg- and WT vs UI.  line 177: "...in the comparison of La-WT and La-arg-....indicated a distinct modulation of arginine and proline metabolis during L. amazonensis infection..." - is this a comparison of the two parasite strains vs UI, or are you comparing results of the two? line 183--185: ...L.amazonensis infected and uninfected macrophages. Enrichment ....from La-WT-infected) and arginase knockout..." - so which is it? Fig. 4: A: why are three metabolites in red font, others in black? B: it would be helpful if the metabolites in B were all in the model in A Line 226: spelling of auxotrofic Lines 240-1: Cat1 and Cat2 increase relative to what? UI? Line 245: "...broking the tricarboxylic acid cycle" - broking? Fig 5:  what do the red up/down arrows represent in the figures?  if they mean something then using two different colors would facilitate interpretation (eg, red down, blue up) do they refer to Fig 4B? If so, there are some disagreements in data such as citrulline

Author Response

Response to reviewer

I reviewed a previous submission of this manuscript. This resubmission is highly improved, with the authors taking care to address virtually all concerns I had regarding the original submission. They have introduced numerous clarifications as well as been more precise in their language. A number of issues still remain but they are minor and should be addressable by the authors. All in all, much improved.

Response: We thank the comments made by the reviewer #1 and we tried to clarify all the points raised, to make the manuscript clear.

Specifics:

p5, lines 133-140: authors state that statistical analyses identified 65 metabolites undergoing changes; I count 67 metabolites in Table 1 (but conversely, there are 65 in Fig 2 heatmap of the data from Table 2?!).

Response: Thank you for the important comment. Following your observation, we counted the significant metabolites and we deleted an erroneous one (adduct mass 281.1119). The final number is 65. We apologize for this mistake.

Table 1: Clarify what column "% change" and VIP mean - former should note what the comparison is made between (specifically, is the infected or uninfected the numerator or denominator?), latter should be spelled out/defined in Table description.  Meaning of upward arrows in "% change" column should be defined in table description.

Response: Thank you for your suggestion. We clarifed the change (%), VIP, ↑ and ↓ in the table caption

“↑, represents the presence of the metabolite in La-WT-infected macrophages or La-arg--infected macrophages and the absence in the uninfected macrophages; ↓, represents the presence of the metabolite in uninfected macrophages and the absence in infected macrophages. % change, represents the increase (+) or decrease (-) in metabolite abundance in the infected groups, this was calculated as follows (Average [infected macrophage] - Average [uninfected]/(Average [uninfected])x 100. VIP (Variable Importance in Projection) represents the weight of that metabolite in the separation in the OPLS-DA model, and values higher than 1 reflect strong differences of that metabolite between the groups. These values were calculated, by default, from all extracted compound data using SIMCA 14.1-1 software. NS, non-significant”

Table 2: what is the reference of change to: is it the Arg- (nominator) vs WT (denominator)? This should be clarified. I don't see a Non-significant p value, so "NS, non-significant" not required again, what is down arrow mean? Missing two column headers to the right of "VIP" column

Response: We corrected the headers of columns and changed the table description to clarify the information:

“↓, represents the presence of the metabolite in La-arg--infected macrophages compared to La-WT-infected macrophages. % change, represents the increase (+) or decrease (-) in metabolite abundance in the infected groups, this was calculated as follows (Average [BALB/c-La-arg—]-Average [BALB/c-La-WT]/ (Average [BALB/c-La-WT])x 100. VIP (Variable Importance in Projection) represents the weight of that metabolite in the separation in the OPLS-DA model, and values higher than 1 reflect strong differences of that metabolite between the groups. These values were calculated, by default, from all extracted compound data using SIMCA 14.1-1 software.”

p11, Line 171: trehalose appears to be misspelled

Response: We correct the spelling in the text.

Line 177-185: this section is unclear on whether the comparison is Arg- vs WT, or Arg- and WT vs UI. 

Response: We correct in the text to clarify the information: “metabolites peak areas from La-WT- infected compared to La-arg--infected macrophages (Figure 3, Table S1), La-WT-infected compared to uninfected macrophages (Figure S3, Table S2) and La-arg--infected macrophages compared to uninfected macrophages (Figure S4, Table S3).”

Line 177: "...in the comparison of La-WT and La-arg-....indicated a distinct modulation of arginine and proline metabolis during L. amazonensis infection..." - is this a comparison of the two parasite strains vs UI, or are you comparing results of the two?

Response: We correct in the text to clarify the information: “… in the comparison of La-arg--infected versus La-WT-infected macrophages, ...”

line 183--185: ...L. amazonensis infected and uninfected macrophages. Enrichment ....from La-WT-infected) and arginase knockout..." - so which is it?

Response: We correct the information about comparison between La-arg--infected macrophages versus La-WT-infected macrophages

Fig. 4: A: why are three metabolites in red font, others in black? B: it would be helpful if the metabolites in B were all in the model in A

Response: In Fig 4A, we represent the urea cycle, showing metabolites from L-arginine metabolism providing the polyamines and trypathione production, and correlates metabolites from host and parasite. Metabolites represented in black color are present in macrophages and parasite. Metabolites in red are exclusively found in parasite.

In Fig 4B, on trial to contemplate the metabolite present in the Fig 4A,we included the graph for peak areas of L-alanyl-aspartate, ϒ-glutamyl-L-cysteine and glycine

Line 226: spelling of auxotrofic

Response: We correct the spelling.

Lines 240-1: Cat1 and Cat2 increase relative to what? UI?

Response: We corrected in the text to explain that is compared to uninfected macrophages

Line 245: "...broking the tricarboxylic acid cycle" - broking? Fig 5:  what do the red up/down arrows represent in the figures?  if they mean something then using two different colors would facilitate interpretation (eg, red down, blue up) do they refer to Fig 4B? If so, there are some disagreements in data such as citrulline

Response: We change the colors of arrows to clarify the representation of increase and decrease amount of metabolite: Blue arrows represent the higher levels of metabolites found in La-WT-infected-macrophages compared to uninfected (upper) or La-arg--infected-macrophages compared to uninfected (down).
